# Anti-Myocardial Infarction Effects of Radix Aconiti Lateralis Preparata Extracts and Their Influence on Small Molecules in the Heart Using Matrix-Assisted Laser Desorption/Ionization–Mass Spectrometry Imaging

**DOI:** 10.3390/ijms20194837

**Published:** 2019-09-29

**Authors:** Hao Wu, Xi Liu, Ze-yu Gao, Zhen-feng Dai, Ming Lin, Fang Tian, Xin Zhao, Yi Sun, Xiao-ping Pu

**Affiliations:** 1National Key Research Laboratory of Natural and Biomimetic Drugs, Peking University, Beijing 100191, China; 2Department of Molecular and Cellular Pharmacology, School of Pharmaceutical Sciences, Peking University, Beijing 100191, China

**Keywords:** Radix Aconiti Lateralis Preparata, myocardial infarction, left anterior descending artery ligation, matrix-assisted laser desorption/ionization–mass spectrometry imaging, energy metabolism, phospholipid

## Abstract

Radix Aconiti Lateralis Preparata (fuzi) is the processed product of *Aconitum carmichaelii* Debeaux tuber, and has great potential anti-myocardial infarction effects, including improving myocardial damage and energy metabolism in rats. However, the effects of Radix Aconiti Lateralis Preparata extracts in a rat model of myocardial infarction have not yet been fully illustrated. Herein, Radix Aconiti Lateral Preparata was used to prepare Radix Aconiti Lateralis Preparata extract (RAE), fuzi polysaccharides (FPS), and fuzi total alkaloid (FTA). Then, we aimed to compare the effects of RAE, FPS, and FTA in MI rats and further explore their influence on small molecules in the heart. We reported that Radix Aconiti Lateralis Preparata extract (RAE) and fuzi total alkaloid (FTA) significantly improved left ventricular function and structure, and reduced myocardial damage and infarct size in rats with myocardial infarction by the left anterior descending artery ligation. In contrast, fuzi polysaccharides (FPS) was less effective than RAE and FTA, indicating that alkaloids might play a major role in the treatment of myocardial infarction. Moreover, via matrix-assisted laser desorption/ionization–mass spectrometry imaging (MALDI–MSI), we further showed that RAE and FTA containing alkaloids as the main common components regulated myocardial energy metabolism-related molecules and phospholipids levels and distribution patterns against myocardial infarction. In particular, it was FTA, not RAE, that could also regulate potassium ions and glutamine to play a cardioprotective role in myocardial infarction, which revealed that an appropriate dose of alkaloids generated more obvious cardiotonic effects. These findings together suggested that Radix Aconiti Lateralis Preparata extracts containing an appropriate dose of alkaloids as its main pharmacological active components exerted protective effects against myocardial infarction by improving myocardial energy metabolism abnormalities and changing phospholipids levels and distribution patterns to stabilize the cardiomyocyte membrane structure. Thus, RAE and FTA extracted from Radix Aconiti Lateralis Preparata are potential candidates for the treatment of myocardial infarction.

## 1. Introduction

Myocardial infarction (MI) following myocardial ischemia is a severe cardiovascular disease which can be caused by coronary artery occlusion [1]. MI occurs when the heart cannot efficiently pump enough oxygen-rich blood to circulate throughout the body, resulting in an insufficient energy supply [1,2]. Over time, the heart cannot tolerate this stress, and the cardiomyocytes become dysfunctional and damaged, eventually leading to MI [3,4]. Cardiomyocyte apoptosis and necrosis play important roles in MI pathogenesis [5]. Clinically, patients experience persistent and severe pain, with progressive changes in their electrocardiograms that can further induce arrhythmias, heart failure or shock [2,6]. Therefore, MI should be prevented and treated as soon as possible. Drugs targeting MI are attracting increasing attention, and thrombolytic agents, beta-blockers (e.g., metoprolol (METO)), analgesics, angiotensin-converting enzyme inhibitors, and statins have become common treatments for MI [7]. Previous studies have confirmed that beta-blockers (such as METO, etc.) can effectively treat heart disease, including MI [8].

The preparation of MI in small animals by left coronary artery ligation was discovered by Johns and Olson in 1954 [9]. After many improvements made by researchers, such as Smith and colleagues, it has become a common model for the study of MI [10,11]. In this study, a modified Smith method was performed by left anterior descending artery (LAD) ligation to develop a rat model of ST-segment elevation MI, which can well simulate the pathological characteristics of clinical myocardial ischemia-induced MI [12,13].

Radix Aconiti Lateralis Preparata (fuzi), the processed product of *Aconitum carmichaelii* Debeaux tuber, family Ranunculaceae, has been widely applied for many years in China and other Asian countries [14]. Evidence has revealed that Radix Aconiti Lateralis Preparata primarily contains alkaloids and polysaccharides to exert beneficial cardioprotective effects, including improved inotropic effect and left ventricular diastolic function [15], improving energy metabolism [16], scavenging hydroxyl radicals and inhibiting lipid peroxidation [17,18] in rats. In addition, aconite tuber (fuzi) increases NO production in humans to improve peripheral blood circulation [19]. These reports suggest the great potential of Radix Aconiti Lateralis Preparata to treat MI. However, there has not been a systematic study on the effects of Radix Aconiti Lateralis Preparata extracts in a rat model of MI, and their influences on heart metabolism-related small molecules. In the present study, Radix Aconiti Lateral Preparata was used to prepare Radix Aconiti Lateralis Preparata extract (RAE), fuzi polysaccharides (FPS), and fuzi total alkaloid (FTA). Then, we aimed to compare the effects of RAE, FPS, and FTA in MI rats and further explore their influence on small molecules in the heart by using matrix-assisted laser desorption/ionization–mass spectrometry imaging (MALDI–MSI).

MALDI–MSI allows the simultaneous in situ detection of the levels and spatial distribution patterns of small molecules in cryosectioned tissues [20]. The technique is simple, rapid, and high-throughput with high accuracy. Using 1,5-diaminonaphthalene hydrochloride as a matrix, MALDI–MSI can detect energy metabolism-related molecules, phospholipids, metal ions, and amino acids in heart tissues [21], all of which are important for studying the roles of Radix Aconiti Lateralis Preparata extracts in MI rats.

## 2. Results

### 2.1. Standardization of Radix Aconiti Lateralis Preparata Extracts

To determine major chemical components of Radix Aconiti Lateralis Preparata extract (RAE), fuzi polysaccharides (FPS), and fuzi total alkaloid (FTA), the peaks for benzoylmesaconine, benzoylaconine, benzoylhypacoitine, and hypaconitine were identified in the HPLC chromatograms as shown in Figure 1. The levels of these four components were 2.58, 0.63, 1.52, and 0.0552 mg/g, respectively, in RAE and 5.78, 1.41, 3.27, and 0.0925 mg/g, respectively, in FTA. FPS did not contain these alkaloids. In addition, polysaccharide levels in RAE, FPS, and FTA were 6.87% (68.66 mg/g), 15.53% (155.3 mg/g), and 0, respectively. We found that alkaloids were the major common components of RAE and FTA, while polysaccharides were the major common components of RAE and FPS.

### 2.2. RAE, FPS, and FTA Improve the Hemodynamic Status and Organ Weight Index of Rats with Myocardial Infarction

Subsequently, we confirmed the protective effects of RAE, FPS, and FTA on cardiac function of MI rats. Results of hemodynamic analysis indicating alterations in left ventricular end-diastolic pressure (LVEDP) and heart rate are shown in Figure 2. LVEDP was inversely correlated with left ventricular diastolic function; moreover, LVEDP was obviously increased in the MI model group (*p* < 0.001), but treatment with 0.4, 0.8, and 1.6 g/kg RAE; 0.4 and 1.6 g/kg FPS or 0.8 and 1.6 g/kg FTA significantly reduced LVEDP (*p* < 0.001, *p* < 0.001, *p* < 0.01, *p* < 0.001, *p* < 0.01, *p* < 0.01, and *p* < 0.01, respectively), indicating that all three extracts improved left ventricular function in MI rats. Moreover, metoprolol reduced LVEDP in the rats with MI (*p* < 0.001). Thereafter, there was no significant difference in heart rate amongst the groups, consistent with prior findings [11].

Persistent ventricular dysfunction can cause abnormality of myocardial structure in MI rats, so we examined changes of organ weight index in rats as shown in Figure 3. Compared with the sham surgery (S) group, the MI model group showed a significant decrease in body weight (*p* < 0.001) but increase in heart weight index (HWI) and lung weight index (LWI) (both *p* < 0.001). Treatment with 0.8 and 1.6 g/kg RAE, 1.6 g/kg FPS or 0.8 and 1.6 g/kg FTA reduced HWI (*p* < 0.05, *p* < 0.05, *p* < 0.05, *p* < 0.05, and *p* < 0.01, respectively); treatment with 0.8 and 1.6 g/kg RAE, 0.4 and 1.6 g/kg FPS or 0.8 g/kg FTA reduced LWI (*p* < 0.05, *p* < 0.05, *p* < 0.01, *p* < 0.05, and *p* < 0.05, respectively). In addition, treatment with metoprolol significantly improved HWI and LWI in the rats with MI (*p* < 0.01 and *p* < 0.01, respectively). These results demonstrated that RAE, FPS, and FTA can stabilize cardiac function and improve structural abnormalities.

### 2.3. RAE, FPS, and FTA Inhibit Myocardial Injury of Rats with Myocardial Infarction

Further, we explored the effect of RAE, FPS, and FTA on myocardial injury of MI rats through staining in cardiac sections. As shown in Figure 4, hematoxylin–eosin (HE) staining indicated severely damaged myocardial fibers (green arrow), obvious vascular lesions (red arrow), and inflammatory cell infiltration (lilac arrow) in the MI model group compared with the S group. Treatment with metoprolol, 0.8 and 1.6 g/kg RAE, 1.6 g/kg FPS or 0.8 and 1.6 g/kg FTA reduced the extent of myocardial fibers damage, vascular lesions, and inflammatory cell infiltration. Particularly, treatment with metoprolol, 0.8 g/kg RAE or 0.8 g/kg FTA produced more evident cardiotonic effect.

In addition, as shown in Figure 5A, in the MI model group, Masson staining showed many blue collagen fibers in the infarct zone in the left ventricles (red arrow), with a thinner left ventricular wall; compensatory hypertrophy was observed in the right ventricle (green arrow). Treatment with RAE, FPS, or FTA improved ventricular hypertrophy. Moreover, treatment with 0.8 g/kg RAE or 0.8 g/kg FTA significantly reduced infarct size in MI rats (both *p* < 0.01) (Figure 5B). Furthermore, metoprolol significantly decreased the size of the myocardial tissue lesions (*p* < 0.05) (Figure 5B). These results showed that RAE, FPS, and FTA could improve myocardial damage, while RAE and FTA were more effective, especially 0.8 g/kg RAE and 0.8 g/kg FTA, indicating that alkaloids as the main common components might play a major role in the treatment of MI. Therefore, 0.8 g/kg RAE and 0.8 g/kg FTA groups were selected to further explore the effects of Radix Aconiti Lateralis Preparata extracts on MI rats via MALDI–MSI.

### 2.4. RAE and FTA Regulate Energy Metabolism-Related Molecules

Considering that insufficient myocardial energy supply is an important factor for MI development, the therapeutic effects of Radix Aconiti Lateralis Preparata might be related to energy metabolism. In fact, previous studies have shown that Radix Aconiti Lateralis Preparata improves myocardial energy metabolism [22]. Therefore, we investigated the effect of RAE and FTA on energy metabolism-related molecules. Based on Masson-stained images of the tissue sections, the infarct zone was located and marked with red circles in MALDI–MSI images (Figure 6). Compared with the S group, the MI model group showed significantly increased xanthine and glucose levels (both *p* < 0.05) and significantly decreased creatine, AMP and ADP levels (*p* < 0.001, *p* < 0.01, and *p* < 0.05, respectively) in the whole transverse sections of heart tissues of MI rats. In contrast, compared with the MI model group, the RAE-treated group showed increased GMP and ADP levels (both *p* < 0.01), whereas the FTA-treated group showed reduced glucose levels (*p* < 0.05) but increased creatine, AMP, GMP, ADP, and ATP levels (*p* < 0.05, *p* < 0.001, *p* < 0.01, *p* < 0.001, *p* < 0.05, respectively) (Figure 6B–H). In addition, in the MI model group, xanthine and glucose in the heart were primarily distributed in the infarct zone, whereas creatine, AMP and ADP were distributed in the non-infarct zone. RAE and FTA treatments predominately altered the distribution patterns of these molecules in the non-infarct zone. Therefore, we can confirm that RAE and FTA can improve myocardial energy supply against MI.

### 2.5. RAE and FTA Decrease Phospholipids

Since the cardiac structure is associated to cardiac function, dysfunctional myocardial energy metabolism might arise due to impaired cardiomyocyte and mitochondrial membrane structures. As the important components of the membrane, phosphatidic acid (PA), phosphatidylethanolamine (PE), and phosphatidylinositol (PI) are involved in cellular and mitochondrial functions [23,24]. As shown in Figure 7, the infarct zone was located and marked like Figure 6A. In the S group, PA and PI were maintained at low overall cellular levels; however, the MI model group showed increased PA (18:0/20:4), PA (18:0/22:6), and PI (18:0/20:4) levels in the whole transverse sections of heart tissues (*p* < 0.01, *p* < 0.01, and *p* < 0.01, respectively), indicating that PA and PI are closely associated with MI. Interestingly, only PE (16:0/18:1) levels amongst the seven phospholipids were reduced in the MI model group (*p* < 0.05). In fact, as the second most abundant phospholipid in the mitochondrial membrane, PE is rapidly synthesized in the mitochondria [24]. Considering that MI impairs mitochondrial function, PE (16:0/18:1) appears to be more closely related to mitochondrial abnormality than PE (18:0) and PE (P-18:0/22:6). Compared with the MI model group, RAE and FTA treatment reversed the dysfunctional changes in PE (18:0), PA (18:0/20:4), and PE (P-18:0/22:6) levels (all *p* < 0.05), further indicating that RAE and FTA can stabilize myocardial membrane structure.

### 2.6. FTA Changes Potassium Ions and Glutamine

In addition, given that potassium ions and amino acids are both important cardioprotective agents in the heart [25,26,27], the levels and distribution patterns of the two molecules were detected to verify the anti-MI effect of FTA and RAE. Compared with the S group, the MI model group showed significantly increased potassium ion and aspartate levels in the whole transverse sections of heart tissues (*p* < 0.01 and *p* < 0.05, respectively) (Figure 8A,B); glutamine and glutamate levels significantly decreased in the MI model group (*p* < 0.001 and *p* < 0.05, respectively) (Figure 8C,D). Compared with the MI model group, the FTA-treated group reduced potassium ion levels (*p* < 0.05) and increased glutamine levels (*p* < 0.05), while the RAE-treated group showed no significant changes. In addition, in the MI model group, potassium ions were evenly distributed in the infarct and non-infarct zones (located and marked like Figure 6A), and glutamine was primarily distributed in the non-infarct zone. FTA treatment altered the distribution patterns of potassium ions and glutamine, particularly in the non-infarct zone. 

All these together strongly suggest that RAE and FTA protected against myocardial injury by improving myocardial energy metabolism abnormalities, altering phospholipids levels and distribution patterns to stabilize the cardiomyocyte membrane structure, and regulating potassium ions and glutamine.

## 3. Discussion

Radix Aconiti Lateralis Preparata as the processed product of *Aconitum carmichaelii* Debeaux tuber exhibits a series of pharmacological functions, including cardiotonic, anti-inflammatory, and antioxidant properties [28]. In addition, *Aconitum* alkaloids, which mainly include diester-diterpene alkaloids (DDAs, mainly include aconitine, mesaconitine, and hypaconitine) and monoester-diterpene alkaloids (MDAs, mainly comprise benzoylmesaconitine, benzoylaconitine, and benzoylhypacoitine), are responsible for the pharmacological activities of Radix Aconiti Lateralis Preparata [29]. However, *Aconitum* alkaloids also have toxicity to the cardiovascular system. As the major toxic components, DDAs are easily hydrolyzed to generate corresponding MDAs in the presence of water or heat, reducing the toxicity to 1/200–1/500 of that of DDAs [30,31], chemical structures of which were shown in Figure 1E. In the present study, on the basis of the Chinese Pharmacopoeia 2015 Edition (2015), Wang [32], and Luo [33], we used Radix Aconiti Lateralis Preparata to prepare low-toxic RAE, FPS, and FTA via a detailed and reproducible process. Component analysis demonstrated that FTA mainly contained benzoylmesaconitine, benzoylaconitine, benzoylhypacoitine, and a quite low concentration of hypaconitine without polysaccharides. In contrast, FPS mainly contained polysaccharides without alkaloids. Moreover, the quality standards of the three extracts met the requirements of the Chinese Pharmacopoeia 2015 Edition (2015) and the previous studies [30,32,33,34].

Next, we evaluated and compared the effects of RAE, FPS, and FTA in a rat model of MI induced via LAD ligation. The present study demonstrated that RAE and FTA significantly ameliorated myocardial injury, improved cardiac function and inhibited cardiac hypertrophy and pulmonary congestion following MI. Particularly, treatment with 0.8 g/kg RAE or 0.8 g/kg FTA produced the most evident cardiotonic effects. In contrast, FPS was less effective than RAE or FTA. Therefore, we selected 0.8 g/kg RAE and 0.8 g/kg FTA to further explore the anti-MI effects of Radix Aconiti Lateralis Preparata extracts via MALDI–MSI.

Since MI causes mitochondrial dysfunction, we explored changes in the tricarboxylic acid cycle- and energy metabolism-related small molecules in the heart. Based on MALDI–MSI findings, obviously abnormal alterations in xanthine, creatine, glucose, AMP, GMP, ADP, and ATP levels indicated that myocardial mitochondrial tricarboxylic acid cycle and energy metabolism was indeed disrupted, even was destroyed. Moreover, the specific distribution patterns of these molecules suggested that MI led to abnormal glucose metabolism, resulting in reduced levels and altered distribution patterns of downstream molecules in the infarct zones, which indicated severe energy supply disorders mainly appeared in the infarct zones. In addition, previous studies have shown increased plasma xanthine oxidase activity and uric acid concentration in acute MI rats, while the change in xanthine, which can be converted to uric acid by the action of xanthine oxidase, in the heart were not specified [35]. In addition, this study confirmed specific increase in xanthine levels in the infarct zone, probably demonstrating decreased myocardial xanthine oxidase activity. After treatments of RAE and FTA, energy supply in cardiomyocytes was increased by altering the levels and distribution patterns of these molecules, which was predominately observed in the non-infarct zones. The decrease in glucose level indicates that FTA improved the compensatory effect of cardiomyocytes in the non-infarct zones after MI, which was related to the reduction of myocardial damage, thereby increasing the levels of AMP, GMP, ADP, and ATP in the heart. Moreover, ADP and ATP were more clearly distributed around the infarct zones to increase the energy supply of cardiomyocytes in the infarct zones and reduce myocardial death, which delayed the occurrence of MI. Furthermore, increase of glutamine level in the hearts of FTA-treated group can indirectly affect ATP homeostasis and energy supply in cardiomyocytes [25,26,36]. These results together confirmed that RAE and FTA containing alkaloids as the main common components exerted anti-MI effects by regulating energy metabolism.

To further verify the anti-MI activity, we investigated the effects of RAE and FTA on phospholipids in the heart. Previous studies have shown that cardiac infarct zones exhibit apoptosis and scar formation and that non-infarct zones exhibit cardiac hypertrophy [37]. The observed distribution patterns of phospholipids in the infarct and non-infarct zones in our rat model of MI might be closely related to myocardial remodeling within the corresponding areas. In particular, phospholipids with a 20:4 fatty acyl chain were predominately distributed in the infarct zone, whereas phospholipids with a 22:6 fatty acyl chain tended to concentrate in the non-infarct zones, suggesting that PA-, PE- and PI-related pathways play important roles in myocardial remodeling. In addition, these phospholipids have many unsaturated fatty acids, which are predisposed to conversion to oxidized phospholipids by reactive oxygen species due to abnormal energy metabolism. Oxidized phospholipids are associated with many inflammatory diseases, such as metabolic disorders, etc. [38]. Therefore, RAE and FTA treatment likely reverse the dysfunctional changes in phospholipids to stabilize the membrane structure and improve myocardial remodeling; indirectly affect the overproduction of oxidized phospholipids and inhibit the occurrence of oxidative damage. Moreover, a previous study has indicated that defects in PA-mediated signaling pathways might represent a novel mechanism of cardiac dysfunction in congestive heart failure due to MI [39]. Although our study can provide useful insights through the distribution of PA, further investigation is needed. Furthermore, increase in the levels of cardiac potassium ion in MI rats was reversed by FTA, perhaps as a result of the stability of potassium channels or the sodium–potassium pump via altering phospholipids on the cardiomyocyte membrane.

Therefore, these findings together should support that RAE and FTA improved myocardial energy metabolism abnormalities and alter phospholipids levels to stabilize the cardiac membrane structure, suggesting that alkaloids might be the main pharmacological active component for MI treatment since Radix Aconiti Lateralis Preparata primarily contained alkaloids and polysaccharides. Notably, 0.8 g/kg FTA treatment appeared to be more effective than 0.8 g/kg RAE treatment in improving the abnormal alterations in small molecules, which might be attributed to higher levels of alkaloids in FTA. Considering that the efficacy of 1.6 g/kg FTA treatment in improving myocardial ischemia was not superior to that of 0.8 g/kg FTA, we speculate that an appropriate dose of alkaloids in Radix Aconiti Lateralis Preparata extracts produced more obvious cardiotonic effects.

The associations amongst the anomalous small molecules are summarized in Figure 9. Mitochondrial energy metabolism was impaired partly because of aberrant changes in phospholipids, which led to the alterations in energy metabolism-related small molecules and amino acids. At the same time, dysfunctional phospholipids on the cell membrane likely disrupted the stability of potassium ion channels or sodium–potassium pumps, resulting in alterations in potassium ion. Thus, Radix Aconiti Lateralis Preparata extracts containing alkaloids as its main active component exhibited anti-MI effects by altering the levels and distribution patterns of several associated molecules.

In conclusion, Radix Aconiti Lateralis Preparata extracts can improve myocardial damage, enhance myocardial function and decrease HWI and LWI in a rat model of MI induced via LAD ligation. The MALDI–MSI results indicated that Radix Aconiti Lateralis Preparata extracts with an appropriate dose of alkaloids reversed the anomalous alterations in the levels and distribution patterns of energy metabolism-related molecules, phospholipids, potassium ions, and glutamine in the heart, thereby improving myocardial energy metabolism abnormalities and altering the phospholipids levels and distribution patterns to produce anti-MI effects. Thus, RAE and FTA extracted from Radix Aconiti Lateralis Preparata are potential candidates for MI treatment.

## 4. Materials and Methods

### 4.1. Reagents and Drugs

Heparin sodium was purchased from Beijing Suolaibao Technology Co., Ltd. (Beijing, China). 1,5-Diaminonaphthalene was purchased from Sigma-Aldrich (St. Louis, MO, USA). d-(+)-glucose (250 mg) was purchased from Shanghai Macklin Biochemical Technology Co., Ltd. (Shanghai, China). Hematoxylin, eosin, Ponceau and aniline blue were provided by Servicebio (Wuhan, China).

Radix Aconiti Lateralis Preparata (batch no. 151002) was purchased from Sichuan Jiangyou Zhongba Aconiti Lateralis Radix Technology Development Co., Ltd. (Sichuan, China). 

Radix Aconiti Lateralis Preparata extracts were processed on the basis of the Chinese Pharmacopoeia 2015 Edition (2015), Wang [32], and Luo [33], as shown in Appendix A. A total of 6000 g of Radix Aconiti Lateralis Preparata was weighed, soaked, and boiled in 72 L of purified water twice, 2 h each time (at least 1 h after boiling), and the suspensions were combined, filtered, and concentrated by the distilling apparatus. The suspension was treated at 4 °C for 24 h and filtered. Then the obtained filtrate was boiled at 115 °C for 1 h and filtered to obtain liquid Radix Aconiti Lateralis Preparata extract. The liquid extract was concentrated to 2 L and divided into two portions: one portion was concentrated and dried to obtain 208.1 g RAE; the other portion was treated with 80% ethanol at 4 °C for 24 h, followed by centrifugation at 4000 rpm for 20 min to get filtrate and precipitate. The filtrate was concentrated and dried to obtain 58.3 g FTA. The precipitate was dissolved in 1 L of water and deproteinized three times with the same vol of 5:1 CHCl_3_–n-BuOH by the Sevag method. The resulting aqueous fraction was concentrated to 500 mL, then 30% hydrogen peroxide solution was added, and the pH was adjusted to 8–9 with 0.15 mol/L sodium hydroxide solution, followed by a 45 °C water bath for 4 h. Then, the suspension was extensively dialyzed against double-distilled water for 3 days and precipitated by addition of 5 vol of 80% ethanol at 4 °C for 24 h. After centrifugation the precipitate was washed with absolute ethanol, acetone, and diethyl ether three times, and then dried in a vacuum at 50 °C to obtain 106.7 g FPS. RAE, FPS, and FTA were all stored at 4 °C and dissolved in normal saline to prepare suspensions of corresponding concentrations before administration.

### 4.2. Preparation of Alkaloid Reference Substances

For high-performance liquid chromatography (HPLC) analysis, benzoylmesaconine (production batch no. 111795-201701), benzoylaconine (production batch no. 111794-201701) and benzoylhypacoitine (production batch no. 111796-201701), mesaconitine (production batch no. 120065-201701), hypaconitine (production batch no. 120066-201701), and aconitine (production batch no. 120067-201701) were purchased from the National Institutes for Food and Drug Control (Beijing, China). Stock solutions of the three reference substances were prepared in methanol and stored at 4 °C. Before analyses, the stock solutions were diluted to obtain standard solutions of benzoylmesaconine (9.7781 μg/mL), benzoylaconine (10.0461 μg/mL), benzoylhypacoitine (10.0074 μg/mL), mesaconitine (5.1466 μg/mL), hypaconitine (5.0148 μg/mL), and aconitine (4.9993 μg/mL).

### 4.3. Quantitative Analysis of Alkaloids

Alkaloids were chromatographically analyzed by the Agilent 1100 HPLC system (Thermo Fisher Scientific, Waltham, MA, USA) using a Thermo Hypersil GOLD C18 column (4.6 mm × 250 mm, 5.0 μm) at a column temperature of 30 °C. The mobile phases (A: 0.2% triethylamine (pH adjusted to 5.3 with glacial acetic acid) and B: CH_3_CN) were 70% A for 0–30 min, 45% A for 30–35 min. The flow rate was maintained at 1.0 mL/min, and the detection wavelength for HPLC analysis was set at 235 nm. The levels were calculated according to the external standard method.

### 4.4. Quantitative Analysis of Polysaccharides

Glucose reference substance (10 mg) was accurately weighed and dissolved in purified water to make up a final volume of 100 mL, and then to be formulated into standard solutions of concentration gradients, followed by adding 1.0 mL of 5% phenol solution and 6.0 mL of concentrated sulfuric acid respectively. Each solution was mixed uniformly and placed at room temperature for 5 min. After being heated in boiling water for 15 min, the absorbance of each standard solution was measured at 484 nm using a UV-visible spectrophotometer (Shimadzu UV-2600, Kyoto-fu, Japan) to establish a regression equation. In addition, sample solutions were prepared, and the absorbance was measured. Polysaccharide levels in three samples were then calculated using the regression equation and conversion coefficients.

### 4.5. Animals Care

In total, 180 male Sprague Dawley rats weighing 230–260 g and aged 7–8 weeks were purchased from Beijing Hua Fu Kang Biotechnology Co., Ltd. (Beijing, China; licence no. SCXK (Beijing) 2014-0004). The rats were labelled, weighed, and caged, water and food provided ad libitum. Rats were maintained at 22–24 °C and 50%–60% humidity under a 12-h light/dark cycle for 1 week to acclimatize them before surgery. All animal experiments were approved by the Peking University Biomedical Ethics Committee (Beijing, China; approval no. LA2017282; approval date: 28/02/2017) and conducted by experimenters who hold the employment certificate of the Department of Laboratory Animal Science, Peking University Health Science Center, China. All efforts were made to minimize animal suffering.

### 4.6. Surgical Procedures

Preparation of the MI rat model was performed using the modified Smith method. Briefly, after administering intraperitoneal anesthesia using 0.5% pentobarbital sodium (1 mL/100 g), the rats were placed in the supine position and provided positive-pressure ventilation. The limbs were connected to electrocardiogram electrodes, and the electrocardiograms were recorded (Appendix A). Subsequently, thoracotomy was performed to expose the heart, and LAD was threaded and permanently ligated to develop the MI model. The obvious elevation in the ST-segment on the electrocardiogram indicated that the procedure was successful (Appendix A). Surgery in the sham surgery (S) group was conducted without the ligation of LAD.

### 4.7. Grouping and Administration

Except for 17 rats that died during surgery, 100% (9/9) and 77% (118/154) of the rats in the Sham surgery (S) and ligation groups, respectively, survived after 24 h of surgery (Appendix A). The LAD ligation rats were randomly divided into eleven groups: MI model group (MOD); metoprolol-treated group (METO, 1.5 mg/kg); Radix Aconiti Lateralis Preparata extract groups (RAE1, RAE2, and RAE3 receiving RAE at doses of 1.6, 0.8, and 0.4 g/kg, respectively); fuzi polysaccharides groups (FPS1, FPS2, and FPS3 receiving FPS at doses of 1.6, 0.8, and 0.4 g/kg, respectively); fuzi total alkaloid groups (FTA1, FTA2, and FTA3 receiving FTA at doses of 1.6, 0.8, and 0.4 g/kg, respectively). The total numbers of each group were 9, 11, 10, 11, 11, 9, 12, 11, 10, 11, 9, and 13, respectively, while the survival numbers before sacrifice were 9, 8, 8, 9, 9, 7, 9, 8, 7, 8, 8, and 10, for the sham surgery, model, METO, RAE1, RAE2, RAE3, FPS1, FPS2, FPS3, FTA1, FTA2, and FTA3 group, respectively (Appendix A, Appendix A). 

Dosage design basis: the rat dosage of metoprolol used was set at 1.5 mg/kg according to preview studies [40,41]. Metoprolol is a selective β1 receptor blocker used in the clinical treatment of myocardial infarction, the efficacy and safety of which for myocardial infarction has been confirmed [40,42,43]. In addition, the dose of aconitine in Radix Acouiti Lateralis Preparata extracts was better to be controlled at about 0.118 mg/kg [34], so the dose of hypaconitine which is close to the toxicity of aconitine should also be controlled at about 0.118 mg/kg. According to the results in Figure 1, we calculated that the dose of RAE and FTA should be around 2.14 g/kg and 1.28 g/kg, respectively. In order to unify the dose, the high dose of RAE, FPS, or FTA was set at 1.6 g/kg, and the medium and low doses were set at 0.8 g/kg and 0.4 g/kg, respectively. 

Each treatment group was intragastrically administered the corresponding drugs at 10 mL/kg/d, whereas the S and MOD groups were administered normal saline for 2 weeks. The behavior of all rats was subsequently observed and recorded. Compared with the sham surgery group, those in the MOD group had reduced consumption of food and water and decreased activity. After treatment, the rats fed more food and water and increased activity. 

### 4.8. Hemodynamics

After 2 weeks of drug administration, the rats were anesthetized with 0.5% pentobarbital sodium (1 mL/100 g) and the right common carotid artery was connected to a biosignal acquisition and processing system through an arterial catheter to measure left ventricular end-diastolic pressure (LVEDP) and heart rate [11]. 

### 4.9. Histopathological Examination

Following hemodynamic testing, the heart of each rat from each group was quickly excised and fixed in 4% paraformaldehyde for hematoxylin–eosin (HE) and Masson staining. After observing the hearts under a microscope, the infarct size (IS) was calculated using the following formula:

IS (%) = midline infarct length/left ventricular midline circumference × 100 [3].

In addition, the hearts and lungs of all rats were weighed to calculate the heart weight index (HWI) and lung weight index (LWI) using the following formula:

Organ weight index = organ wet weight (mg)/body weight of the rat (g).

### 4.10. Matrix-Assisted Laser Desorption/Ionization–Mass Spectrometry Imaging (MALDI–MSI)

After hemodynamic testing, three rats each from the S, MOD, RAE, and FTA groups were euthanized by intraperitoneal injection of a three-fold anesthetic dose of 0.5% pentobarbital sodium (3 mL/100 g), and the hearts were quickly excised, snap frozen in liquid nitrogen and stored at −80 °C.

Complete and smooth transverse slices of the frozen hearts (thickness, 10 μm) were obtained using a cryostat microtome (Scotsman Jencons, Germany) at −17 °C. The distance from the slice position to the apex was two-fifths of the vertical length of the heart. The tissue slices were then transferred onto indium tin oxide-coated glass slides (Bruker Daltonics, Bremen, Germany) and desiccated in a vacuum pump for 30 min before matrix spraying. Next, the tissue slices were sprayed with an ImagePrep tissue imaging matrix sprayer (Bruker Daltonics, Billerica, MA, USA). The matrix solution was prepared as follows: 39.5 mg of 1,5-diaminonaphthalene was added to a solution of 500 µL of 1 M hydrochloric acid and 4 mL of deionized water. The mixture was ultrasonically dissolved until the visible particles disappeared. High purity anhydrous ethanol (4.5 mL) were then added. The matrix was sprayed according to the following settings: spraying for 2 s, incubation for 20 s, and drying for 75 s, with 10 spraying cycles. After the end of a spray, the slide was sprayed a second time in the opposite direction.

Slices were analyzed by MALDI–MSI using an Autoflex Speed™ MALDI TOF (TOF) system with a 2-kHz Smartbeam-II laser (Bruker Daltonics, Germany) according to the protocols used in previous studies [21,44]. The mass spectra data were acquired over a mass range of *m/z* 80–1000 Da and the imaging spatial resolution was set to 200 µm. The results of MALDI–MSI were analyzed according to the study by Liu and colleagues [44].

### 4.11. Statistical Analysis

Data are expressed as mean ± standard error of the mean. Statistical analyses of MALDI–MSI were performed using SCiLS Lab based on the normalization of total ion chromatography data. All data were analyzed using GraphPad Prism 6.0 (GraphPad Prism, CA, USA). All results were compared using one-way analysis of variance (ANOVA). Comparison between the groups was performed using Fisher’s LSD test. *p* < 0.05 was considered significant.

## Figures and Tables

**Figure 1 ijms-20-04837-f001:**
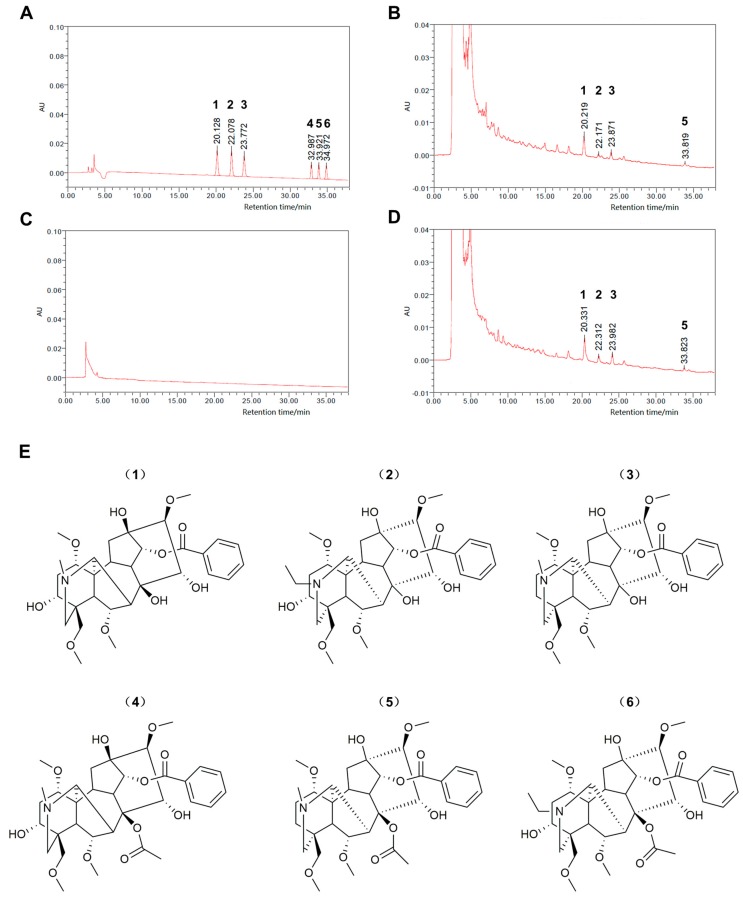
Standardization analysis of Radix Aconiti Lateralis Preparata extracts. HPLC chromatogram of (**A**) reference substances, (**B**) Radix Aconiti Lateralis Preparata extract, (**C**) fuzi polysaccharides, and (**D**) fuzi total alkaloid samples. (**E**) Chemical structure of the six reference substances. **1**, benzoylmesaconine; **2**, benzoylaconine; **3**, benzoylhypacoitine; **4**, mesaconitine; **5**, hypaconitine; **6**, aconitine.

**Figure 2 ijms-20-04837-f002:**
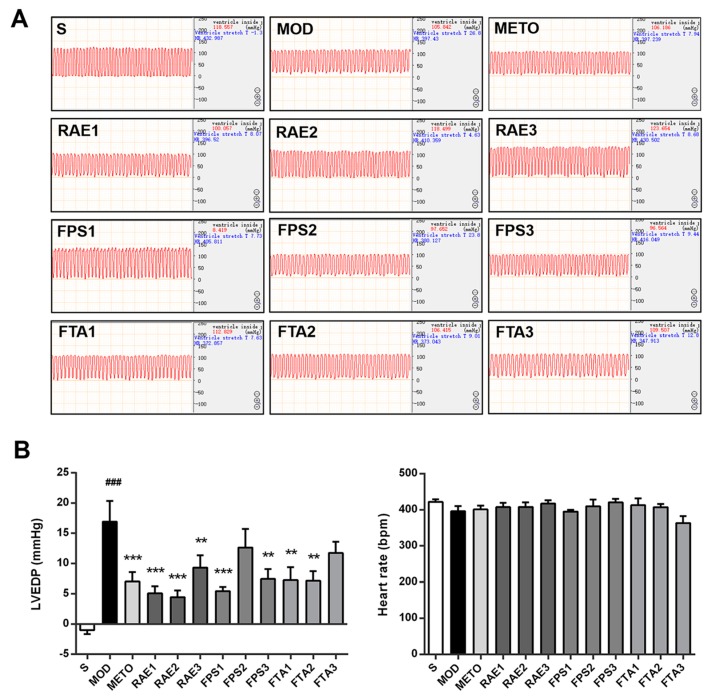
Radix Aconiti Lateralis Preparata extract (RAE), fuzi polysaccharides (FPS), and fuzi total alkaloid (FTA) improved the hemodynamic status of rats with myocardial infarction. (**A**) Hemodynamic diagrams of the hearts of rats in each group. (**B**) Left ventricular end-diastolic pressure (LVEDP) and heart rate. S, sham surgery group; MOD, MI model group; METO, metoprolol-treated group; RAE1/2/3, 1.6/0.8/0.4 g/kg Radix Aconiti Lateralis Preparata extract groups; FPS1/2/3, 1.6/0.8/0.4 g/kg fuzi polysaccharides groups; FTA1/2/3, 1.6/0.8/0.4 g/kg fuzi total alkaloid groups. Data are presented as mean ± standard error of the mean; *n* = 5–7 per group. ^###^
*p* < 0.001 vs. S group; ** *p* < 0.01, *** *p* < 0.001 vs. MOD group.

**Figure 3 ijms-20-04837-f003:**
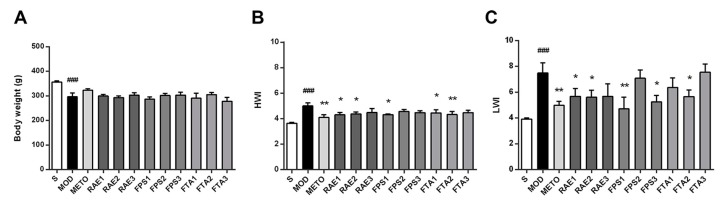
RAE, FPS, and FTA improved organ weight index of rats with myocardial infarction. (**A**) Body weight. (**B**) Heart weight index (HWI). (**C**) Lung weight index (LWI). S, sham surgery group; MOD, MI model group; METO, metoprolol-treated group; RAE1/2/3, 1.6/0.8/0.4 g/kg Radix Aconiti Lateralis Preparata extract groups; FPS1/2/3, 1.6/0.8/0.4 g/kg fuzi polysaccharides groups; FTA1/2/3, 1.6/0.8/0.4 g/kg fuzi total alkaloid groups. Data are presented as mean ± standard error of the mean; n = 5–7 per group. ^###^
*p* < 0.001 vs. S group; * *p* < 0.05, ** *p* < 0.01 vs. MOD group.

**Figure 4 ijms-20-04837-f004:**
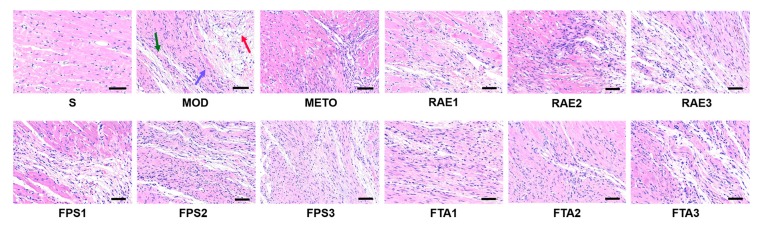
Photomicrographs of heart sections stained with hematoxylin–eosin in each group (close to the coronary vein), scale bar = 60 μm. S, sham surgery group; MOD, MI model group; METO, metoprolol-treated group; RAE1/2/3, 1.6/0.8/0.4 g/kg, Radix Aconiti Lateralis Preparata extract groups; FPS1/2/3, 1.6/0.8/0.4 g/kg fuzi polysaccharides groups; FTA1/2/3, 1.6/0.8/0.4 g/kg fuzi total alkaloid groups. The green arrow indicates damaged myocardial fibers, the red arrow indicates vascular lesions, and the lilac arrow indicates inflammatory cell infiltration.

**Figure 5 ijms-20-04837-f005:**
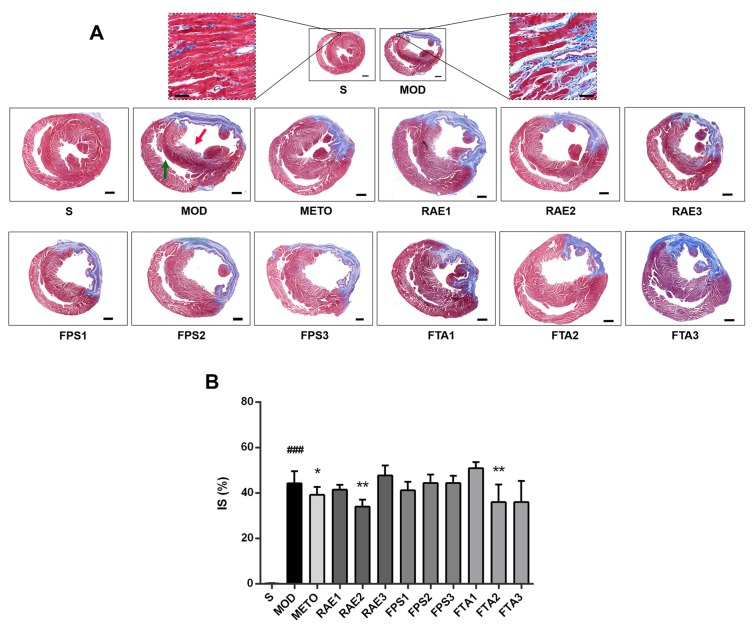
Photomicrographs of heart sections stained with Masson in each group (close to the coronary vein). (**A**) Magnifications of 7.1× in cross-sectional photomicrographs of heart sections stained with Masson in each group, scale bar = 1000 μm; magnifications of 200× in local enlarged photomicrographs near the coronary vein, scale bar = 60 μm. The red arrow indicates the left ventricle, and the green arrow indicates the right ventricle. (**B**) Statistical analyses of infarct size (IS) using Image-Pro Plus. S, sham surgery group; MOD, MI model group; METO, metoprolol-treated group; RAE1/2/3, 1.6/0.8/0.4 g/kg Radix Aconiti Lateralis Preparata extract groups; FPS1/2/3, 1.6/0.8/0.4 g/kg fuzi polysaccharides groups; FTA1/2/3, 1.6/0.8/0.4 g/kg fuzi total alkaloid groups. Data are presented as mean ± standard error of the mean; n = 4–6 per group. ^###^
*p* < 0.001 vs. S group; * *p* < 0.01, ** *p* < 0.01 vs. MOD group.

**Figure 6 ijms-20-04837-f006:**
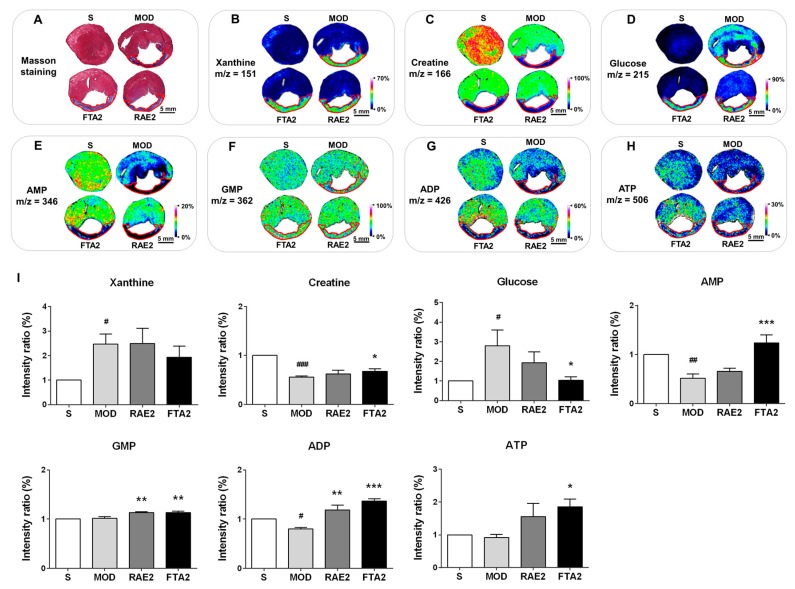
RAE and FTA regulated energy metabolism-related molecules. (**A**) Masson staining of adjacent heart slices. In situ matrix-assisted laser desorption/ionization–mass spectrometry imaging (MALDI–MSI) of (**B**) glucose, (**C**) GMP, (**D**) ATP, (**E**) AMP, (**F**) ADP, (**G**) xanthine, and (**H**) creatine. (**I**) Relative intensity ratios of these molecules were normalized and analyzed using SCiLS Lab. Spatial resolution = 200 μm; scale bar = 5 mm. *m*/*z*: mass-to-charge ratio. S, sham surgery group; MOD, MI model group; RAE2, 0.8 g/kg Radix Aconiti Lateralis Preparata extract group; FTA2, 0.8 g/kg fuzi total alkaloid group. Data are presented as mean ± standard error of the mean; *n* = 3 per group. ^#^
*p* < 0.05, ^##^
*p* < 0.01, ^###^
*p* < 0.001 vs. S group; * *p* < 0.05, ** *p* < 0.01, *** *p* < 0.001 vs. MOD group.

**Figure 7 ijms-20-04837-f007:**
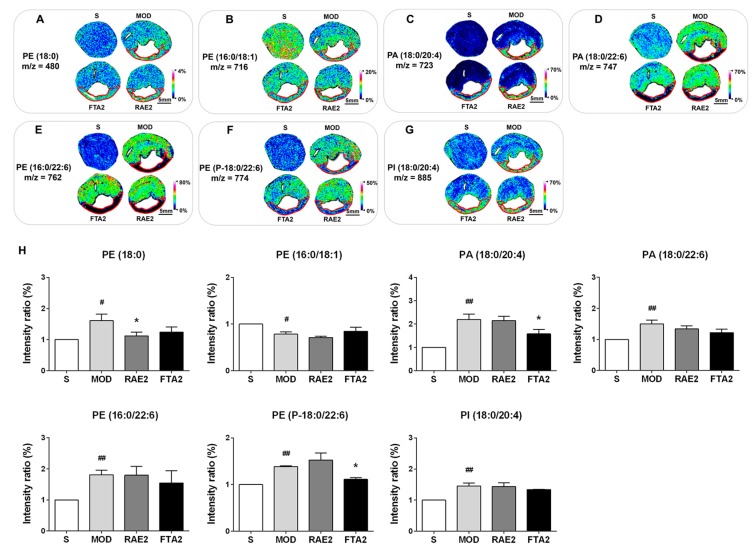
RAE and FTA decreased phospholipids. In situ MALDI–MSI of (**A**) phosphatidylethanolamine (PE) (18:0), (**B**) PE (16:0/18:1), (**C**) phosphatidic acid (PA) (18:0/20:4), (**D**) PE (18:0/22:6), (**E**) PE (16:0/22:6), (**F**) PA (P-18:0/22:6), and (**G**) phosphatidylinositol (PI) (18:0/20:4). (**H**) Relative intensity ratios of the seven phospholipids were normalized and analyzed using SCiLS Lab. Spatial resolution = 200 μm; scale bar = 5 mm. *m/z*: mass-to-charge ratio. S, sham surgery group; MOD, MI model group; RAE2, 0.8 g/kg Radix Aconiti Lateralis Preparata extract group; FTA2, 0.8 g/kg fuzi total alkaloid group. Data are presented as mean ± standard error of the mean; n = 3 per group. ^#^
*p* < 0.05, ^##^
*p* < 0.01 vs. S group, * *p* < 0.05 vs. MOD group.

**Figure 8 ijms-20-04837-f008:**
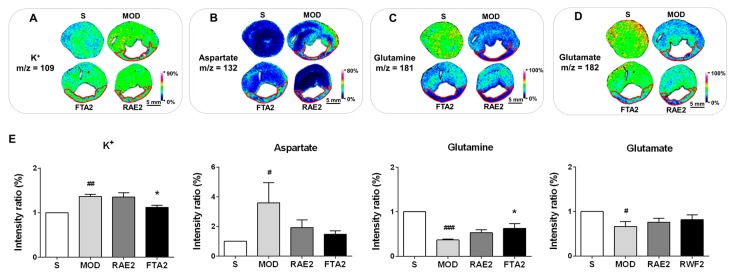
FTA changed potassium ion levels and glutamine. In situ MALDI–MSI of (**A**) potassium ions, (**B**) aspartate, (**C**) glutamine, and (**D**) glutamate. (**E**) Relative intensity ratios of these molecules were normalized and analyzed using SCiLS Lab. Spatial resolution = 200 μm; scale bar = 5 mm. *m/z*: mass-to-charge ratio. S, sham surgery group; MOD, MI model group; RAE2, 0.8 g/kg Radix Aconiti Lateralis Preparata extract group; FTA2, 0.8 g/kg fuzi total alkaloid group. Data are presented as mean ± standard error of the mean; n = 3 per group. ^#^
*p* < 0.05, ^##^
*p* < 0.01, ^###^
*p* < 0.001 vs. S group, * *p* < 0.05 vs. MOD group.

**Figure 9 ijms-20-04837-f009:**
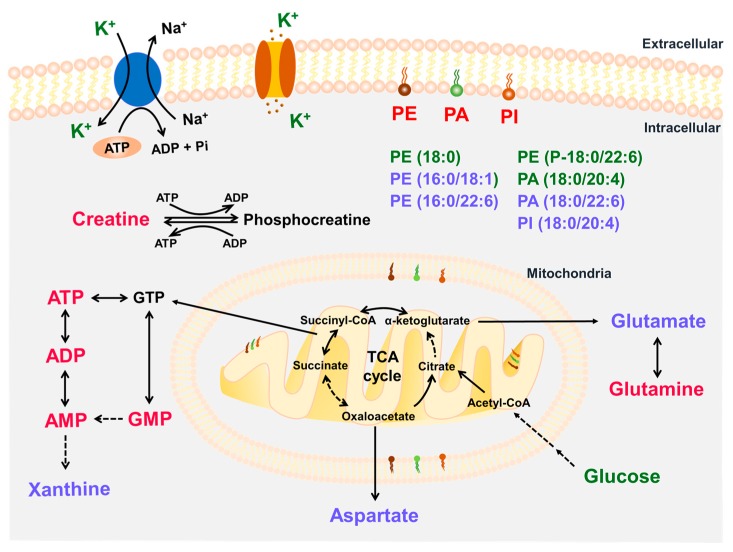
Summary of associations amongst the anomalous small molecules in this study. Small molecules that were significantly increased after treatment with RAE or FTA are represented in red type, and those that were significantly decreased are represented in green type. Those that were obviously altered only in the MOD group are represented in lilac type. Black arrows indicate metabolic processes; dashed arrows indicate the omission of intermediate metabolites.

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
