# Peer review of "Anti-Myocardial Infarction Effects of Radix Aconiti Lateralis Preparata Extracts and Their Influence on Small Molecules in the Heart Using Matrix-Assisted Laser Desorption/Ionization–Mass Spectrometry Imaging"

_ijms, 2019, doi:10.3390/ijms20194837_

Round 1

Reviewer 1 Report

This manuscript by Wu at al. reported results of the effect of Radix Aconti Lateralis preparata (fuzi), fuzi polysacharides and fuzi total alkaloids on damage after myocardial infarction.

Specific comments are provided below for authors' consideration.  

Introduction: Aim or hypothesis of the study is misssing

Methods: need to be improve. Some sub-chapter are in real detail and some are written poorly.

What is model group? Authors are using this name for group, however, no explanation for it.

The duration of ligation and duration of reperfusion should be provided.

Was the ligation reversible?

How did authors euthanize the animals?

Figures:

All abbreviations must be explained in figure legend.

Discussion: it is more recapitulation than real discussion. Findings need to be discuss more with possible pathways included.

Author Response

Response to Reviewer 1 Comments

Point 1: Introduction: Aim or hypothesis of the study is missing.

Response 1Thank you. The sentence in page 2 has been revised.  “Then, we aimed to compare the effects of RAE, FPS, and FTA in MI rats and further explore their influence on small molecules in the heart by using Matrix-assisted laser desorption/ionisation–mass spectrometry imaging (MALDI–MSI)” was added and marked in red (P2, paragraph 3, line 12-14).

Point 2: Methods: need to be improve. Some sub-chapter are in real detail and some are written poorly.

Response 2Thank you. The word “permanently” was added in sub-chapter 4.6; Supplementary Table S1 was added in sub-chapter 4.7;“Three rats each from the S, MOD, RAE and FTA groups were euthanized by intraperitoneal injection of a 3-fold anesthetic dose of 0.5% pentobarbital sodium (3 mL/100 g)” was added in sub-chapter 4.10; “Comparison between the groups was performed using Fisher’s LSD test” was added in sub-chapter 4.11.

Point 3: What is model group? Authors are using this name for group, however, no explanation for it.

Response 3Thank you. Model group indicated the MI model group receiving LAD ligation. In the manuscript, “model group” has been replaced by “MI model group” (P15, paragraph 2, line 3).

Point 4: The duration of ligation and duration of reperfusion should be provided.

Response 4Thank you. The surgery of MI didn’t contain the reperfusion process after LAD ligation, and it takes two weeks to be MI in this study (P15, paragraph 4, line 2).

Point 5: Was the ligation reversible?

Response 5The ligation was not reversible. “LAD was threaded and permanently ligated to develop the MI model” was added and marked in red (P15, paragraph 1, line 5-6).

Point 6: How did authors euthanize the animals?

Response 6: “Three rats each from the S, MOD, RAE and FTA groups were euthanized by intraperitoneal injection of a 3-fold anesthetic dose of 0.5% pentobarbital sodium (3 mL/100 g)” was added and marked in red (P16, paragraph 1, line 1-3).

Point 7: Figures: All abbreviations must be explained in figure legend.

Response 7Thank you. HWI, LWI, IS, PA, PE and PI have been explained in figure legends (P4-10).

Point 8: Discussion: it is more recapitulation than real discussion. Findings need to be discussed more with possible pathways included.

Response 8Thank you. “Since MI causes mitochondrial dysfunction, we explored changes in the tricarboxylic acid cycle- and energy metabolism-related small molecules in the heart. Based on MALDI–MSI findings, obviously abnormal alterations in xanthine, creatine, glucose, AMP, GMP, ADP and ATP levels indicated that myocardial mitochondrial tricarboxylic acid cycle and energy metabolism was indeed disrupted, even was destroyed. Moreover, the specific distribution patterns of these molecules suggested that MI led to abnormal glucose metabolism, resulting in reduced levels and altered distribution patterns of downstream molecules in the infarct zones, which indicated severe energy supply disorders mainly appeared in the infarct zones. In addition, previous studies have shown increased plasma xanthine oxidase activity and uric acid concentration in acute MI rats, while the change in xanthine, which can be converted to uric acid by the action of xanthine oxidase, in the heart were not specified [35]. And this study confirmed specific increase in xanthine levels in the infarct zone, probably demonstrating decreased myocardial xanthine oxidase activity. After treatments of RAE and FTA, energy supply in cardiomyocytes was increased by altering the levels and distribution patterns of these molecules, which was predominately observed in the non-infarct zones. The decrease in glucose level indicates that FTA improved the compensatory effect of cardiomyocytes in the non-infarct zones after MI, which was related to the reduction of myocardial damage, thereby increasing the levels of AMP, GMP, ADP and ATP in the heart. Moreover, ADP and ATP were more clearly distributed around the infarct zones to increase the energy supply of cardiomyocytes in the infarct zones and reduce myocardial death, which delayed the occurrence of MI.” was added in discussion (P11-12, paragraph 3 in discussion).

“In addition, these phospholipids have many unsaturated fatty acids, which are predisposed to conversion to oxidized phospholipids by reactive oxygen species due to abnormal energy metabolism. Oxidized phospholipids are associated with many inflammatory diseases, such as metabolic disorders, etc. [38]. Therefore, RAE and FTA treatment likely reverse the dysfunctional changes in phospholipids to stabilize the membrane structure and improve myocardial remodelling; indirectly affect the overproduction of oxidized phospholipids and inhibit the occurrence of oxidative damage. Moreover, a previous study has indicated that defects in PA-mediated signalling pathways might represent a novel mechanism of cardiac dysfunction in congestive heart failure due to MI [39]. Although our study can provide useful insights through the distribution of PA, further investigation is needed.” Was added in discussion (P12, paragraph 4 in discussion).

Reviewer 2 Report

The main interest of this study was to investigate whether Radix Aconiti Lateralis Preparata extracts supplementation to rats with myocardial infarction can influence abnormalities of myocardial energy metabolism and phospholipids in the MI injured heart tissue. The study is well written and sound.
Experiments were performed carefully and statistics appear correct.
However, some parts should be done more carefully to fully appreciate the work.
1. The abstract should be rephrased. It is described what was performed, but the hypothesis or the main purpose of the study is missing.
2. METO group is explained the first time in part 4.7. It will be clearer to mention something about metoprolol in the introduction part.
3. The discussion part is not very discussed with other studies.
4. Part 4.7 is written very opaquely. It will be better to have data about animal grouping and extracts administration in the table
5. Figure 1E- “chemical structure of the substances” is not necessary. Figure 1E is not also mentioned in the text.
6. Some shortcuts are not defined in the text (etc. LVEDP, HR….)
7. In the figures is changed RAE1 to PAE1
8. Pictures in Figure 4 should be better described and not only in the MOD group. In METO group is not bar
9. 3 animals in the group are not adequate
10. Post hoc test in statistical analysis is missing

Round 2

Reviewer 2 Report

Thank you very much to authors, the article was improved and I recommend to accept it in this form.